# Interactive Explanations of Agent Behavior

## Abstract

As reinforcement learning methods increasingly amass accomplishments, the need for comprehending their solutions becomes more crucial. Most explainable reinforcement learning (XRL) methods generate a static explanation depicting their developers' intuition of what should be explained and how. In contrast, literature from the social sciences proposes that meaningful explanations are structured as a dialog between the explainer and the explainee, suggesting a more active role for the user and her communication with the agent. In this paper, we present ASQ-IT – an interactive tool that presents video clips of the agent acting in its environment based on queries given by the user that describe temporal properties of behaviors of interest. Our approach is based on formal methods: queries in ASQ-IT's user interface map to a fragment of Linear Temporal Logic over finite traces (LTLf), which we developed, and our algorithm for query processing is based on automata theory. We provide experimental results from a user-study aimed at testing ASQ-IT's usability, and report positive outcomes from both objective performance and self-reported ability of participants to use our tool.

## Introduction

Reinforcement Learning (RL) has shown impressive success in recent years; e.g., mastering Go or achieving human-level performance in Atari games (Silver et al. 2016; Mnih et al. 2015). However, current training techniques are complex and rely on implicit goals and indirect feature representations, and thus largely produce black-box agents. In order for such trained agents to be successfully deployed, in particular in safety-critical domains such as healthcare, it is crucial for them to be trustworthy; namely, both developers and users need to understand, predict and assess agents' behavior. This need has led to an abundance of "explainable RL" (XRL) methods (Dazeley, Vamplew, and Cruz 2021) designed to elucidate black-box agents.

Following Miller (2018), we argue in favor of *interactive XRL methods* that proceed as a dialog between the explainer (system) and the explanee (user): the user repeatedly poses queries for the system to answer. Nearly all research in explanation systems make two deterring assumptions regarding this definition. Firstly, that the question does not arrive

from the explainee, i.e. the user does not construct the query, rather the researchers designed the explanation through their perceived lens of what a viable question is. Secondly, that the output is static, i.e. a single answer as an explanation, instead of a dialog.

Interactive explanations have recently been perceived as a significant future direction for system intelligibility and enhancing user engagement (Abdul et al. 2018). Increasing evidence also points towards interaction and exploration as means to reduce over-reliance on AI recommendations, which occurs even when explanations are provided (Buçinca, Malaya, and Gajos 2021). Insights from this line of work justify the need to not only strive for improved AI performance and techniques but also increase people's motivation for cognitively engaging with the explanations and the system, as can be achieved by promoting interaction and exploration.

A unique challenge in XRL is the ongoing nature of the agents. Nearly all current XRL methods are state based; namely, they attempt to explain a single decision made at a specific time-point. However, a recent study showed that clinicians, for example, prefer understanding the model as a whole, rather than being provided with explanations for each decision Jacobs et al. (2021).

The approach for global explanations that we follow produces a policy summary in the form of clips of the agent interacting with its environment (Amir, Doshi-Velez, and Sarne 2019). These approaches use different criteria to determine which traces of agent behavior to show, such as state importance (Amir and Amir 2018; Huang et al. 2018), ability to generalize a policy (Huang et al. 2017; Lage et al. 2018) or agent disagreements (Amitai and Amir 2022). It has been shown that using such tools is helpful for users to understand an agent's behavior. Since the users' attention span is very limited, the challenge is choosing which handful of clips to present to the user. We note that all of these methods are static; namely, they do not allow user input and their choice of which clips to present is based on a heuristic.

In this work, we develop an interactive XRL tool that aims to assist users to comprehend an agent in a global manner. Our tool generates clips of the agent interacting with its environment. The user controls which clips will be presented by feeding queries that specify properties of clips of interested. The interaction with the tool resembles a dialogue: the user

enters a query, receives a handful of clips that answer it, the user can then refine her query, and the process continues.

As the name suggests, the main challenge in developing an interactive tool is the interaction with human users (especially laypeople). Indeed, unless constrained, study participants pose vague and informal queries that are hard for a tool to process. A tool's interface must strike the right balance between expressivity and usability. We address these challenges as follows. *i)* We develop a simple logic that can express common properties of clips. Note that clips are ongoing, thus our logic must reason about temporal behaviors. An established logic to reason about such properties is Linear Temporal Logic (Pnueli 1977), and our logic relies on its finite counterpart called LTLf (De Giacomo, De Masellis, and Montali 2014). *ii)* Laypeople cannot be expected to produce logic formulas, thus we develop a simple user interface that maps directly to our logic. *iii)* We assume access to a library of agent execution traces. We develop an efficient automata-based algorithm to search this library for clips that answer a user's query.

Our paper presents the following contributions:
It introduces ASQ-IT, an **A**gent **S**ystem **Q**uries **I**nteractive **T**ool that enables users to describe and generate queries towards an agent and receive answers as explanations-through-demonstration of their behavior. We develop an intermediate temporal logic: queries in the tool's user interface map directly to our logic and in turn, our logic constitutes a fragment of LTLf. Our logic constitutes a *specification pattern* for querying finite traces. To the best of our knowledge, while specification patterns are common for verification purposes (Dwyer, Avrunin, and Corbett 1999), this is the first pattern designed for querying.We present preliminary results from a user-study showing that laypeople, with no training in logic or RL, are able to comprehend and generate meaningful queries to ASQ-IT.

## Related Work

This work relates to two main areas of research, which we discuss in this section: (1) explanations in sequential decision-making settings and (2) interactive explanations.

**Explanations in sequential decision-making settings.** In this paper, we focus on the problem of explaining the behavior of agents operating in sequential decision-making settings. Work in this area is typically concerned with explaining policies learned through Reinforcement Learning.

RL explanation methods can be roughly divided into two classes. *Local* explanations focus on explaining specific agent decisions (Krarup et al. 2019; Khan et al. 2011; Hayes and Shah 2017; Booth, Muise, and Shah 2019; Anderson et al. 2020), e.g., by showing what information a game-playing agent attends to in a specific game state (Greydanus et al. 2017), or generating causal explanations (Madumal et al. 2020). In contrast, *global* explanations aim to convey the agent's policy rather than explain particular decisions. One approach to global explanations is to generate a proxy model of the policy that is more interpretable, e.g., through policy graphs (Topin and Veloso 2019) or decision trees approximating the policy (Coppens et al. 2019). In this

paper, we utilize the idea of extracting demonstrations of agent behavior as a global explanation (Amir, Doshi-Velez, and Sarne 2019) to answer queries posed by users, such that they can interactively explore the agent's policy and its characteristics.

**Interactive Explanations** Some early works on decision-support systems provided users with interactive explanation methods. For example, MYCIN (Davis, Buchanan, and Shortliffe 1977), a system for clinical decision-support, allowed its users to pose "why" and "how" questions and responded by revealing the rules that led to a particular inference. Such explanations are more difficult to provide in current systems that do not use a logic-based representation. Few works in interpretable machine learning also designed interactive explanations for supervised learning models. For instance, TCAV is a method that enables users to test whether the model relies on a user-determined concept in its decision-making (Kim et al. 2018). Recently, this approach has been applied to analyzing the chess knowledge of AlphaZero (McGrath et al. 2021). Interactive XRL has been flagged as a promising research direction in interactive RL research (Arzate Cruz and Igarashi 2020) Most closely related to the problem we discuss are the works of Hayes and Shah (2017) and Cruz and Igarashi (2021) both of which introduce systems to help their users debug agent behavior through interactive interfaces. Both works shape the user's interaction through a limited set of action-related questions such as "when a particular action will be taken?" or "why wasn't an alternative action chosen?", while our focus seeks to bestow more freedom for expressivity and exploration.

## Preliminaries

We first describe the domain used in this work so as to provide a running example.

**Highway Domain.** The domain, shown in Figure 1, consists of a busy highway with numerous cars, represented as blue rectangles, moving along multiple highway lanes (numbered from top to bottom). Cars can accelerate, decelerate, and change lanes. The agent controls and navigates a green car. As there are no defined targets in this domain, we can observe the agent's general behavior and preferences instead of focusing on its progression towards some goal. The behavior of the agent is determined by its reward function (e.g., getting rewards for driving fast, driving in the right lane, etc.) and its training process.

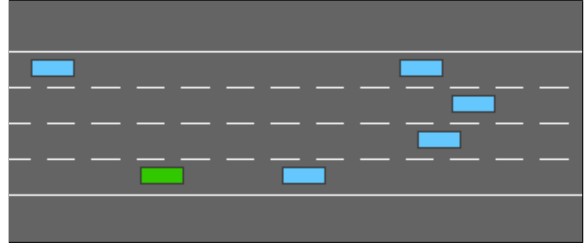

Figure 1: Highway domain.

**Markov Decision Process** For the purpose of this work we assume an MDP setting, formally defined as a tuple $\langle S, A, Tr, R \rangle$ where $S$ is the set of states, $A$ the set of actions, $R$ the reward function mapping states to numerical values, and $Tr : S \times A \times S$ the transition function mapping transitions between two states via an action to a probability. A solution to an MDP is a *policy* $\pi$.

**Abstracting the MDP** Consider an MDP $\mathcal{A}$ over a state space $S$. A predicate $p$ is a function $p : S \to \{\texttt{True}, \texttt{False}\}$ denoting whether some attribute exists in $s$. For example, given a state $s$ in the highway domain, the predicate $\texttt{lane-1}$ returns $\texttt{True}$ iff the agent (green car) is in Lane 1 in $s$ and the predicate $\texttt{behind}$ returns $\texttt{True}$ iff the agent is driving behind some blue car. Note that there are possibly many states for which a predicate can answer $\texttt{True}$. For example, $\texttt{behind}$ returns $\texttt{True}$ both in a state in which the agent is driving in the top lane and in the bottom lane as long as it is driving behind a blue car.

We assume a domain expert both chooses $P$, and produces a mapping from $S$ to an abstract-state space $\Sigma = 2^P$. the mapping $f : S \to \Sigma$, implies that for a state $s \in S$, the function $f(s)$ returns the set of predicates that are true at $s$, thus $f(s) = \sigma = \{p \in P \mid p(s) = \texttt{True}\}$.

Next, consider an abstract-trace $\eta = \sigma_1 \ldots, \sigma_k \in \Sigma^*$ that answers some query. The tool's output to the user will be any trace $\tau = s_1, \ldots, s_k \in S^*$ that maps to the abstract trace $\eta$, i.e. any $\tau$ such that $f(s_i) = \sigma_i$, for every $1 \le i \le k$.

For example, suppose that $P = \{\texttt{lane-1}, \texttt{behind}\}$. In a state $s_1$ for which $f(s_1) = \{\texttt{lane-1}, \texttt{behind}\}$, necessarily the agent is traveling in Lane 1 *and* behind a blue car. On the other hand, in a state $s_2$ for which $f(s_2) = \{\texttt{lane-1}\}$, the agent is traveling in Lane 1, but since $\texttt{behind}(s_2) = \texttt{False}$, the lane in front of it is necessarily empty.

## ASQ-IT

We develop ASQ-IT, **A**gent **S**ystem **Q**uries **I**nteractive **T**ool, an interactive explainability tool, allowing users to iteratively query an agent regarding its behavior and interaction with its surroundings until achieving their desired level of understanding and trust. The following section describes the three main components of ASQ-IT:

*i) Query interface*: Allows users to express their queries in layperson terms with no coding background required,
*ii) Back-end*: Processes the input query and produces relevant answer, and
*iii) Explanation Interface*: Presents answers to the user's queries in a meaningful and clear manner.
We now provide an overview of the tool's usage for clarity.

**Tool Usage, An Illustration.** The first interaction users have with ASQ-IT is through the Query Interface (Figure 2, left) where they are able to define scenarios and behaviors they wish to observe in the agent's interaction with the environment. To do this, users define a start and end state for the agent using predefined predicates available through drop-down menus, along with constraints on the behaviors they wish to see. Once selected, these specifications construct a query that is passed to the back-end process, which

then searches and retrieves compatible trajectories from a database of the agent's execution traces. These are made into video clips and presented to users in the Explanation Interface (Figure 2, right).

**Example 1.** Say we would like to understand how the agent would go about crossing multiple lanes. We can specify the agent's start state as *Lane 1* and the end state as *Lane 4*. The resulting output would be all the sequences which depict the agent crossing these lanes.

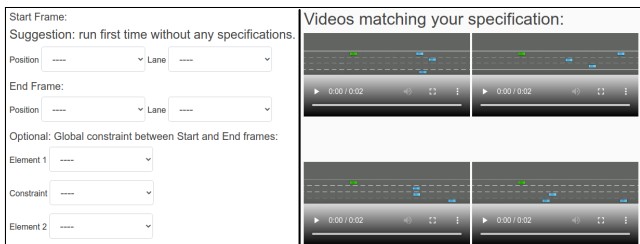

Figure 2: Interactive Explanation Tool - Highway domain. **Left**: Query Interface; **Right**: Explanation Interface.

## A Logic For Expressing Queries

At a high-level, the goal of ASQ-IT is to provide traces that answer a user's query. The tool is based on two components: (1) a language with which users can express the traces that they would like to observe, and (2) an algorithm that finds traces that answer a user's query. In this section we describe the theory behind (1) and the algorithm for (2) is described in the following section. We start by surveying Linear Temporal Logic on Finite Traces (LTLf), on which our query language is based. We then introduce our intermediate logic based on LTLf. Each query in the user interface maps to a formula in our logic.

### Linear Temporal Logic on Finite Traces

An LTLf formula is defined with respect to a set of predicates $P$. It consists of the standard propositional logical operators, i.e., $\wedge$, $\vee$, and $\neg$, together with temporal operators: $X$ read "next", $U$ read "until", $F$ read "eventually", and $G$ read "always". Formally, the syntax of a formula is:

$$\phi \to P \mid \phi \wedge \phi \mid \neg\phi \mid X\phi \mid \phi U \phi$$

We use the temporal abbreviations $F\phi \equiv \texttt{True}U\phi$ and $G\phi \equiv \neg F \neg \phi$ in addition to the standard $\vee$.

**Example 2.** We describe examples of formulas together with their intuitive semantics. Let $P = \{\texttt{lane-1}, \texttt{behind}\}$.

- The formula $G$ $\texttt{lane-1}$ (read "always Lane 1") specifies traces in which the agent travels only in Lane 1.
- The formula $G$ $\neg\texttt{lane-1}$ specifies traces in which the agent is never in Lane 1.
- The formula $X$ $\texttt{lane-1}$ specifies traces in which the agent is driving in Lane 1 in the second position of the trace. Similarly, the formula $X^m$ $\texttt{lane-1}$, for $m \in \mathbb{N}$, is short-hand for a sequence of $m$ $X$ operators, specifies

traces in which the agent is driving in Lane 1 in the $m$-th position of the trace. The formula $X^m$ `True` specifies traces of length at least $m$.

- The formula `lane-1` $U$ `behind` (read "Lane 1 until behind") specifies traces in which initially, the agent drives in Lane 1, and it does so continuously until it is behind some blue car. After that point, there are no restrictions. Note that the trace might contain only one position in which the agent is in Lane 1 and behind a blue car.
- The formula $F$ `lane-1` (read "eventually Lane 1") specifies finite traces in which the agent visits Lane 1 at least once.

We turn to formalize the semantics of LTLf. Consider an LTLf formula $\phi$, which recognizes a set of finite traces over $\Sigma = 2^P$. Consider a trace $\eta = \sigma_1, \ldots, \sigma_k \in \Sigma^*$. The definition of when $\eta$ satisfies $\phi$, denoted $\eta \models \phi$, is done by induction on the structure of $\phi$:

- If $\phi = p \in P$, then $\eta \models \phi$ iff $p \in \sigma_1$.
- If $\phi = \phi_1 \wedge \phi_1$ then $\eta \models \phi$ iff $\eta \models \phi_1$ and $\eta \models \phi_2$.
- If $\phi = \neg\phi_1$ then $\eta \models \phi$ iff $\eta \not\models \phi_1$.
- If $\phi = X\phi_1$ then $\eta \models \phi$ iff $(\sigma_2, \ldots, \sigma_k) \models \phi_1$.
- If $\phi = \phi_1 U \phi_2$ then $\eta \models \phi$ iff there is an index $1 \leq i \leq k$ such that $(\sigma_i, \ldots, \sigma_k) \models \phi_2$ and for each $1 \leq j \leq i$, we have $(\sigma_j, \ldots, \sigma_k) \models \phi_1$.

### The Query Language

Let $P$ be a set of predicates and let $\Sigma = 2^P$. A *simple query* is based on the following components:

- A description of the start and end state of the trace. These are given as propositional formulas $\phi_s$ and $\phi_e$ over the predicates $P$. For example, when $\phi_s = \neg$`lane-1` $\wedge$ `behind`, in any trace returned to the user, in the first position of a trace the green car is not on Lane 1 and behind some car.
- A constraint on the trace between $\phi_s$ and $\phi_e$, which is given as a third propositional formula $\phi_c$ over $P$. We elaborate below on specific constraints.

**Stays constant**  The constraint $\phi_c$ *stays constant* between $\phi$ and $\phi_e$ is written in LTLf as

$$(\phi_s \wedge \phi_c) \bigwedge X(\phi_c U e).$$

For example, let $\phi_e = $ `lane-1`, $\phi_e = $ `lane-4`, and $\phi_c = $ `behind`. Then, the formula represents traces in which initially the agent is driving in Lane 1. The trace ends with the agent driving in Lane 4, and during the whole trace, there is a car behind the agent.

**Changes**  The constraint $\phi_c$ *stays constant* between $\phi_s$ and $\phi_e$ is written in LTLf as

$$(\phi_s \wedge \phi_c) \bigwedge X F(\neg\phi_c \wedge F\phi_e).$$

For example, let $\phi_s = $ `lane-1`, $\phi_e = $ `lane-4`, and $\phi_c = $ `behind`. Then, the formula represents traces in which initially the agent is driving in Lane 1 behind some car. At some point during the trace, the agent is not behind any car. The trace ends with the agent driving in Lane 4.

As a second example, let $\phi_s = \phi_e = \phi_c = $ `lane-1`. This formula represents traces in which initially as well as at the end of the trace, the agent drives in Lane 1. However, the constraint requires there to be a point in which the agent does not drive in Lane 1.

**Changes into**  Let $\phi_c'$ be a second propositional formula over $P$. The constraint $\phi_c$ *changes into* $\phi_c'$ between $\phi_s$ and $\phi_e$ is written in LTLf as

$$(\phi_s \wedge \phi_c \wedge \neg\phi_c') \bigwedge X F(\neg\phi_c \wedge \phi_c' \wedge F\phi_e).$$

For example, let $\phi_c = $ `lane-1`, $\phi_c' = $ `lane-4`, and $\phi_e = $ `lane-1`. Then, the formula represents traces in which initially the agent is driving in Lane 1. At some point in the trace, the agent drives in Lane 4. The trace ends with the agent driving in Lane 1.

### Compositional queries

As we will see in the next section, our algorithm to answer queries takes as input a general LTLf formula. Thus, allowing composition of the queries from the previous section, as long as the composition produces an LTLf formula, comes at no cost.

Consider two queries $\varphi_1$ and $\varphi_2$ in their LTLf form. Applying propositional operators on $\varphi_1$ and $\varphi_2$ results in an LTLf formula. For example, let $\varphi_1$ be the query `lane-2` stays constant and $\varphi_2$ be the query `behind` changes. Then, the query $\varphi_1 \wedge \varphi_2$ asks for traces in which the agent stays in Lane 2, and the situation in front of it changes: at the beginning of the trace it is behind a car and during the trace, the lane empties.

As a second example, often, users like to see long traces since they contain "more information" regarding the agent. This is easily implemented using conjunction. Recall that the query $X^m$ `True` is satisfied by traces of length at least $m$. Consider some query $\varphi$. Then, the query $\varphi \wedge X^m$ `True`, asks for traces of length at least $m$ that answer $\varphi$.

Concatenation is another useful operation. It allows to devise queries that ask for traces that visit "intermediate states" between $s$ and $e$. We describe how to concatenate queries. Let $\varphi_2$ be a query between states $\phi_i$ and $\phi_e$. Consider a second query $\varphi_1$ between $\phi_s$ and $\phi_e'$. Now, replace the instance of $\phi_e'$ in $\varphi_1$ with $\varphi_2$. We call the resulting query $\varphi_1 \cdot \varphi_2$. A trace that answers $\varphi_1 \cdot \varphi_2$ consists of a trace that starts at $\phi_s$, visits the *intermediate state* $\phi_i$, and ends in $\phi_e$, all the while satisfying the required constraints between $\phi_s$ and $\phi_i$ and between $\phi_i$ and $\phi_e$. For example, let $\phi_s = $ `behind` $\wedge$ `lane-2`, $\phi_i = $ `above` $\wedge$ `lane-1`, and $\phi_e = $ `in-front-of` $\wedge$ `lane-2`. A trace in which the agent is in Lane 2 and overtakes a car from above (Lane 1) satisfies this query.

### Query-Answering Algorithm

In this section we describe an algorithm to answer queries. We assume that in a pre-processing step, the domain expert collects simulations of the agent in the domain. For

ease of presentation, we assume that one (long) trace $w = w_1, \ldots, w_n \in S^*$ is collected and its abstraction $\eta = \sigma_1, \ldots, \sigma_n \in \Sigma^*$ is generated. We solve the following problem.

**Problem:** Given an LTLf query $\varphi$, find a sub-trace $\sigma_\ell, \ldots, \sigma_m$ that satisfies $\varphi$.

Before describing the algorithm, we state its guarantees.

**Theorem 3.** *Given an LTLf formula $\varphi$ and a trace $\eta$ of length $n$ over the alphabet $\Sigma$, the algorithm return a sub-trace that satisfies $\varphi$, if one exists. The algorithm processes $\eta$ at most twice, thus its running time is $O(n)$.*[1]

The algorithm is based on automata. For completeness, we survey the required definitions and results.

**Nondeterministic Finite Automata** A nondeterministic automaton (NFA, for short) is a tuple $\mathcal{A} = \langle \Sigma, Q, \delta, Q_0, Acc \rangle$, where $\Sigma$ is an alphabet, $Q$ is a set of states, $\delta : Q \times \Sigma \to 2^Q$ is a transition function, $Q_0 \subseteq Q$ is an initial state, and $Acc \subseteq Q$ is a set of accepting states. A run of $\mathcal{A}$ on a word $w = \sigma_1 \sigma_2 \ldots \sigma_k \in \Sigma^*$ is $r = r_0, r_1, \ldots, r_k \in Q^*$, where $r$ starts in an initial state, i.e., $r_0 \in Q_0$, and respects the transition function, i.e., for each $i \geq 1$, we have $r_i \in \delta(r_{i-1}, \sigma_i)$. We say that $r$ is *accepting* iff it ends in an accepting state, i.e., $r_k \in Acc$. We say that $\mathcal{A}$ *accepts* $w$ if there is a run on $w$ that is accepting. The *language* of $\mathcal{A}$, denoted $L(\mathcal{A})$, is the set of words that it accepts.

Consider an NFA $\mathcal{A} = \langle \Sigma, S, \delta, Q_0, Acc \rangle$. Our algorithm simulates the execution of automata on words. That is, the algorithm feeds letters to $\mathcal{A}$ one at a time while keeping track of which states the automaton can be in. More formally, we think of $\mathcal{A}$ as an object with a field `CurrentStates` and two functions. The function $\mathcal{A}.\text{init}()$ initiates a new run by setting `CurrentStates` $= Q_0$. The function $\mathcal{A}.\text{step}(\sigma)$ reads the letter $\sigma$ by updating the current states to `CurrentStates`$' = \cup_{s \in \texttt{CurrentStates}} \delta(s, \sigma)$. We assume that $\mathcal{A}.\text{step}(\sigma)$ returns `True` iff one of the current states is accepting.

**Lemma 4.** *Suppose that the automaton reads the word $w$ one letter at a time. Then, the last call to $\text{step}$ returns `True` iff $w \in L(\mathcal{A})$.*

Finally, our algorithm is based on a translation between LTLf and NFAs, formally stated as follows.

**Theorem 5.** *(Giacomo and Vardi 2013) Consider an LTLf formula $\varphi$ over a set of predicates $P$. There is an NFA $\mathcal{A}_\varphi$ over the alphabet $\Sigma = 2^P$ whose language is the set of traces that $\varphi$ recognizes. That is, for every trace $\eta \in \Sigma^*$ we have $\eta \in L(\mathcal{A})$ iff $\eta \models \varphi$. The number of states in $\mathcal{A}_\varphi$ is $2^{O(|\varphi|)}$.*

**The algorithm** Consider an LTLf formula $\varphi$ and a trace $\eta = \sigma_1, \ldots, \sigma_n$ over $\Sigma = 2^P$. Our goal is to find a sub-trace of $\eta$ that satisfies $\varphi$.

We describe a first attempt at solving the problem. Generate an NFA $\mathcal{A}_\varphi$ whose language consists of the traces that

---

[1]We assume that $n$, the length of the trace, is much larger than sizes of the queries. For short traces, the running time needs to take into account also the size of the queries.

satisfy $\varphi$. For each index $\ell \geq 1$, decide if there is a sub-trace of $\eta$ starting from position $\ell$ that satisfies $\varphi$ by iteratively feeding the suffix $\sigma_\ell, \sigma_{\ell+1}, \ldots, \sigma_n$ of $\eta$ into $\mathcal{A}_\varphi$. If $\mathcal{A}_\varphi$ accepts at an index $m$, we simply return the sub-trace $\sigma_\ell, \ldots, \sigma_m$ as an answer to the query. This sub-trace is in the language of $\mathcal{A}_\varphi$, thus by Theorem 5, it answers the query $\varphi$.

While this algorithm is correct, it is not efficient: since for each index $\ell$, in the worst case, we read the whole suffix $\sigma_\ell, \ldots, \sigma_n$, the running time of the algorithm is $\Theta(n^2)$. Recall that we assume that $n$ is very large, thus such a high running time will cause significant lag when answering queries.

Our algorithm to answer queries traverses the trace $\eta$ at most twice, thus its running time is $O(n)$. The algorithm operates as follows. In addition to $\mathcal{A}_\varphi$, we obtain an NFA for $F\varphi$ (read "eventually $\varphi$"). We execute $\mathcal{A}_{F\varphi}$ on $\eta$ until we find an index $m$ such that $\mathcal{A}_{F\varphi}$ accepts $\sigma_1, \ldots, \sigma_m$. That is, when $\mathcal{A}_{F\varphi}$ accepts, by Lemma 4, we are guaranteed that the trace has a suffix that satisfies $\varphi$.

We are not done. Since our goal is to find a sub-trace that satisfies $\varphi$, we still need to find an index $\ell$ such that $\sigma_\ell, \ldots, \sigma_m$ satisfies $\varphi$. To do that, we read the trace backwards (starting from $\sigma_m$ and until $\sigma_1$) and execute $\mathcal{A}_\varphi$ "backwards", i.e., starting from the accepting states and accepting once an initial state is reached. Formally, let $\mathcal{A}_\varphi = \langle \Sigma, Q_\varphi, \delta_\varphi, Q_\varphi^0, F_\varphi \rangle$. We initiate the run of $\mathcal{A}_\varphi$ by setting `CurrentStates` $= F_\varphi$. We execute $\mathcal{A}_\varphi$ backwards. Suppose that the letter $\sigma_i$ is read. Then, we update `CurrentStates`$' = \{s \in S_\varphi : \exists s' \in \texttt{CurrentStates} \text{ s.t. } s' \in \delta(s, \sigma_i)\}$. We accept if `CurrentStates` contains an initial state. It is not hard to show that $\sigma_m, \ldots, \sigma_\ell$ is accepted in this manner iff $\sigma_\ell, \ldots, \sigma_m$ satisfies $\varphi$. Moreover, it is clear that $\eta$ is read (forward) once by $\mathcal{A}_{F\varphi}$ and read at most once (backwards) by $\mathcal{A}_\varphi$, thus the running time is linear in $n$.

## Implementation Design

This section describes the implementation of ASQ-IT which utilizes the query language and the algorithm described in the previous sections.

### Query Interface.

The main point of contact between users and ASQ-IT is through the Query interface. We conducted several pilot studies and iteratively revised the design of the query interface. In the first pilot studies, we let users specify open-ended queries in natural language. This step helped reveal the types of questions users were interested in, but was both hard for participants (did not always know what they could ask) and would also make answering queries infeasible. Once we identified the question types and determined the language for queries (specifying start and end states and constraints), we tried several designs for inputting this information. We began with more free-form designs, such as teaching users how to form queries through text-boxes. Finally, in order to constrain the query-space and reduce cognitive load, a restricted drop-down based interface was adopted.

The drop-down menu is designed to clearly and simply

| LTLf → DFA | Search for Traces | Video Generation |
|---|---|---|
| 1.155 ±.153 | **0.191 ±.121** | 3.268 ±.39 |

Table 1: Algorithm component runtime in seconds, averaged over 10 queries varying in complexity.

guide users towards possible state specifications for constructing their queries. Predicates, i.e. state attributes, are grouped into types for reducing cognitive load and to avoid excessive options. For instance, all lane specifications appear under one drop-down, as these are mutually exclusive, due to the agent being only present in one at a time. Initially the interface also included the option to add intermediate states, however, this option was not used by most users and made the interface more complex. Therefore, we removed it from the current interface.

### Back-end

The workings of ASQ-IT's back-end was described in detail in preceding sections. An minor change that was added to the algorithm's implementation for user convenience was a lower and upper bound on the length of the traces retrieved. This was used to prune long traces (videos) that fit the query in terms of start and end state but do not actually reflect the user's intention, due to many different behaviors occurring throughout the trace. It is interesting to note that in practice, the running time of the algorithm to search for traces is negligible compared to the system's external components[2] such as generating videos or producing the automata by LTLf to DFA conversion, as seen in Table 1.

### Explanation interface.

Once the query has been submitted to ASQ-IT and all relevant traces have been retrieved, users are shown up to four [3] of these traces in video format, through the explanation interface (Figure 2). Additionally, users can select to *load more videos* in order to see more of the agent's interaction with the environment. An indicator at the top of the interface displays the number of additional traces that have been found and can be viewed upon request. Currently, the order of videos shown to users is random due to no specific priority being restricted. Ordering the videos by some dedicated heuristic or by user-specified choice is a prospect for future work.

At any point during their interaction with the explanation interface, users may select to return to the query interface to construct new queries or alter their previous ones in order to refine the traces retrieved such as to better fit their intentions.

## Empirical methodology

To evaluate ASQ-IT, we conducted a user-study designed to examine the usability of the tool for a layperson.

*Agent.* The Highway agent was trained for 2000 episodes using double DQN architecture (Hasselt 2010) and rewarded

---

[2]FFmpeg (Tomar 2006), MONA (Klarlund and Møller 2001) LTLf2DFA (Fuggitti 2019)

[3]Configurable parameter.

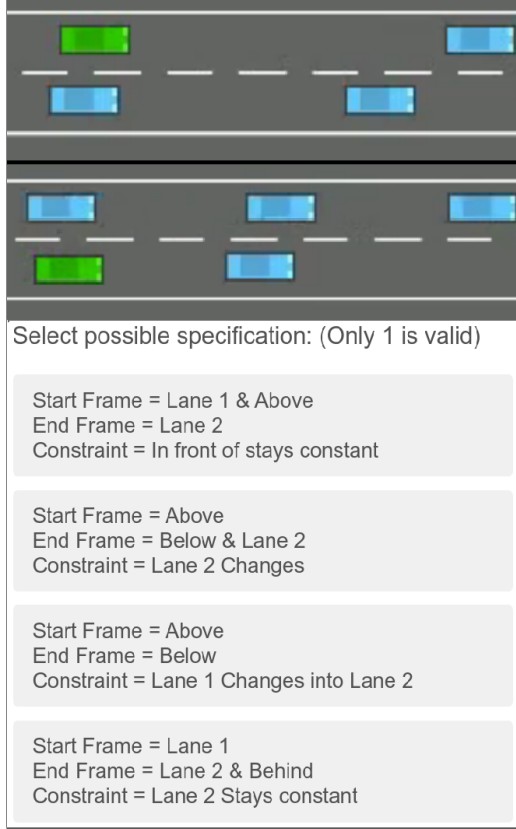

Figure 3: Movie to Query (M2Q) task, question 2.
**Top**: Start State; **Mid**: End State; **Bot**: Multiple Choice.
Full Video: https://bit.ly/3KuHbQf

for avoiding collisions. For the experiment we generated a database of 900 execution traces with varying domain attributes such as number of lanes and car density.

## Experiment

*Hypotheses.* We hypothesize that using our interface, users will be able to quickly grasp the mechanism for generating queries to the agent, thus allowing them to explore its behavior in an interactive fashion. We acknowledge here that a full evaluation of ASQ-IT should include users' benefits from its explanations, and that this work does not yet cover that aspect.

*Participants.* Forty participants were recruited through Prolific (20 female, mean age = 34.7, STD = 11.29), each receiving $4.5 for their completion of the Task. To incentivize participants to make an effort, they were provided a bonus of 15 cents for each correct answer. Participants whose overall task duration were lower than the mean by more than two standard deviations were filtered out.

*Procedure.* First, participants were introduced to the Highway domain and the concept of AI agents. Then commenced an introduction to the ASQ-IT's interface and the process of generating queries for the system. Each explanation was followed by a short quiz to ensure understanding before advancing. As a final step before the task, participants were provided a link to ASQ-IT's interface where

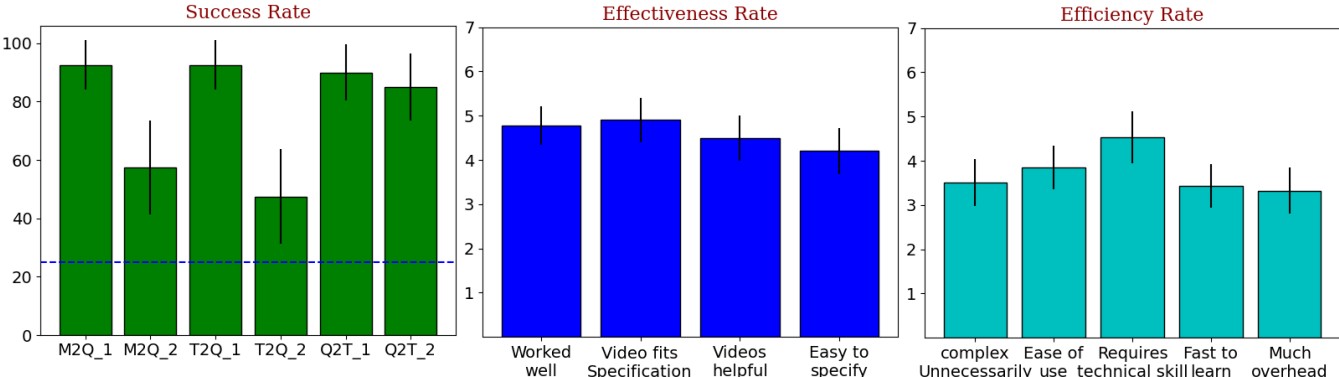

Figure 4: User-Study Results.

they could interact and explore both the interface and the agent. In the task section, participants were tested on their understanding of the interface, query generation and output through three types of tasks. *i) Movies to Queries (M2Q):* Given an output video, select the correct query that would result in its generation (example in Figure 3). *ii) Free Text to Queries (T2Q):* Given textual descriptions of desired behavior, select the correct query. *iii) Queries to Free Text (Q2T):* Given a query, select the correct textual description of the desired behavior. All questions were multiple-choice with four possibilities and a single correct answer and each task type included two questions in ascending difficulty. Figure 3 denotes an example M2Q question[4]. Upon task completion, participants were prompted to provide textual feedback regarding their experience with the system & interface and complete a usability survey.

*Evaluation Metrics and Analyses.* The empirical evaluation consisted of three elements. Firstly, the success rate in answering the task questions. Secondly, the usability survey, based on the system usability scale (Brooke 1996), to measure participants' perception of ASQ-IT in terms of effectiveness & efficiency. Survey questions measured participants' agreement for each item and were rated on a 7-point Likert scale ranging from 1 - "Strongly disagree" to 7 - "Strongly agree". The full study, including all survey items is provided as supplementary. To compare the usability ratings, we averaged the values of the different items normalizing such that higher values always indicate positive judgment of the system. Lastly, as a analysis was done of participants' textual responses regarding their experience in order to extract significant insights or common themes.

## Results

In this section, we describe the user-study results, reporting the main observations made regarding participant experience and use of ASQ-IT.

**Comprehending & Formulating Queries.** We assess participants' ability to understand and to formulate queries based on their performance int the task section of the study,

---

[4]Full user study available at https://bit.ly/3JjfKZy

where participants were tested on inferring the correct query syntax or outcome. We first note that the success rate was higher than a random guess baseline for all questions, as depicted by the dotted line in the left-most plot in Figure 4. Apart from two questions which seemed to be harder for participants, approximately 90% success rate was observed. This supports our hypothesis, that participants are able to meaningfully comprehend and formulate queries in a bidirectional manner, be it by translation to a query or from one.

Upon closer examination, we identified two fundamental causes for incorrect participant answers: *i) Agent relations (position):* Confusing the position of the agent compared to other cars such as mixing "Behind" with "In Front Of" (e.g. is the agent behind another car or is there one behind the agent?), and *i) Misunderstanding constraints:* Participants that were not able to comprehend or grasp the effect or purpose of enacting constraints on the agent's trace and most often chose to ignore these specifications. These alone were responsible for ≈ 90% of all incorrect answers. An example of such a case can be seen in Figure 3 where the start state (top) depicts the agent in lane 1, above a blue car and the end state shows the agent below a blue car, in lane 2. All multiple choice answers have a specification which is plausible given these two states, and only the constraint specification dictates the correct answer.

We were also able to see improvement throughout the task, where participant who struggled with simple questions regarding constraints would manage to solve correctly harder questions that appeared later on in the task. Some participants self-reported that elements of the interface became clearer when asked to answer questions about them. One participant wrote *"I found the instructions quite hard to understand. When a description was provided and you had to complete what you thought was the correct specification, I found this a better way to learn the process."* It should be noted that participants had access to the tool while completing the task section, such that they could keep exploring and learning about it had they chosen to. The tool itself operates in a "Query to Movie (Q2M)" format, such that it could not have been directly used to answer the questions posed in the task.

**Usability.** Overall there was large variation in participants' responses to the usability survey questions. However, several themes and trends emerged from participants' open-ended responses.

*Effectiveness.* Participants found ASQ-IT, on average, more effective than not, in all effectiveness questions (see Figure 4 middle panel). Multiple responses mentioned its usefulness for testing and observing how the agent acts. Others described positively the fact that it was clear to them what videos would be generated by ASQ-IT, so long as the specification was not very complex, and after some initial trial and error phase. Most complaints targeted the many options available and the complexity in understanding the interface, albeit, many participants went on to report that after some exploration, their experience and understanding greatly improved, suggesting that there is a learning curve in using the tool: *"Yes these seem to become clearer the longer I use the system."*

*Efficiency.* Many participant described some level of uncertainty upon initial interaction with the interface, mainly given the lengthy explanations prior to using it. However, the lion's share of participants reported quickly understanding once access to ASQ-IT was given and some exploration with the interface was conducted. When asked what would help them interact with the tool, many participants responded that they would prefer the interface to have fewer options and more visual aid for the existing ones. Needless to say, there is a trade-off between the simplicity of the interface and its expressivity.

**Expressivity.** When asked to describe what features or behaviors were missing or desired for the highway domain, participants mostly requested the ability to control for the agent's speed and distance from other cars, along with the option to specify the positions of other cars and the output video length.

When asked what agent behaviors and situations were of interest to them, specifiable or not using ASQ-IT's current interface, participants mostly referred to observing the agent react to critical situations such as obstacles on the road, lane merges or interaction of other cars such as emergency vehicles or evasion of accelerating or braking cars.

## Discussion and Future Work

With the growing integration of AI systems in sequential decision-making domains, the need for meaningful and engaging explanations is crucial. One method for increasing trust while reducing over-reliance on these systems is through interactive interfaces and explanations.

We designed and developed ASQ-IT – an XRL interactive tool for querying AI agents using user-constructed queries. These are translated to LTLf and used to search agent execution traces for sub-sequences adhering to them. Relevant sequences are then returned as video-clips to the user, acting as explanation-by-demonstration of the agent in the user specified scenarios.

To explore and assess our tool's usability, several user-studies were iteratively constructed and carried-out, and changes to ASQ-IT's interfaces were made in accordance with user feedback we collected. The primary axis on which adjustments were made was the trade-off between interface complexity and query expressivity. By broadening the range of components users could specify, or adding the ability to define intermediate states, we enabled greater manipulation of the domain and more specific agent behavior, i.e., increasing expressivity. However, we noticed that the more options our interface presented, the more criticism our participants displayed towards complexity and cognitive load which were paired with lower success rates in study tasks. Our iterative, feedback-based approach guided our interface design towards a more constrained direction, presenting users only with the dominant domain features to control.

The results of our user-study indicate that, using our current interface design, users are able to learn and use our query based approach in order to explore and investigate deep reinforcement learning agents all the while reporting, on-average, positive ratings for both the effectiveness and efficiency of ASQ-IT.

We intend to keep pursuing additional means for increasing expressivity while balancing the cognitive complexity. Further future work will be dedicated to validating our results through additional domains while also expanding the scope of our studies to test the explanation benefits of using ASQ-IT. Finally, we also intend to keep iteratively updating and improving both the query and explanation interfaces of ASQ-IT through additional user feedback.

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
