# OpenReview forum: "Interactive Explanations of Agent Behavior"
_icaps-conference.org/ICAPS/2022/Workshop/XAIP — XAIP 2022_

### Official Review · Reviewer_nZ9J · 2022-04-27
**The paper presents an interactive tool for querying the agent about a particular behavior, and the interface can also showcase snippets of video clips to demonstrate the behaviors. Overall, I liked the idea of the interface. The interface design is nicely motivated, and the paper is well written. There are certainly some places for improvement (mentioned below), but I think it is an interesting problem and recommend it for acceptance.**

**Rating:** 8
**Confidence:** 5

**Review:**

Feedback for improvement:

- Few comments on evaluation: First, if the study is not comparing or has no independent variable, I would refer to it as a case study. Second, I would recommend comparing the current interface with some baseline or other methods if authors plan to submit for conference/journal publication. I initially hoped for some inference statistics based on the abstract mentioning user study.
- If you have a study design, I would suggest having some inference statistics. Furthermore, the authors acknowledge that a complete evaluation of ASQ-IT requires users' benefits from explanations/videos. I agree it is an essential component for the full picture of how this interface would compare. I would recommend checking some works from Hal Daumé III to compare different frameworks on query design.
- It is not clear from the abstract the exact benefit this new interface would achieve. I recommend using specific results from the evaluation within the abstract.
- I liked the idea of using video to showcase the queried behavior. Although, it is not clear why video modality is used as output from the manuscript. I would suggest providing some rationale for using video modality compared with semantic.
- Additionally, I am curious to know the limitations of using video as output behaviors. I suppose showcasing behavior in applied domains would have some limitations, especially in applied fields such as robotics. For example, how would a robotic arm show its supposed behavior for a query? Therefore, I suggest providing an application domain where ASQ-IT can be actively applied within the introduction section.
- The paper uses the highway domain as a running example to explain sections of methods. I liked the idea of grounding the algorithm in a running example. One suggestion is to use a specific scenario that could be referenced back to a figure. Currently, Figure 1 is not conveying much detail. The authors can easily replace Figure 1 with a specific example that serves as a running example for explaining the algorithm.
- I would also suggest adding an algorithm block to showcase the whole picture of how ASQ-IT operates.

---

### Official Review · Reviewer_xqkp · 2022-04-27
**An interesting approach to retrieve execution traces**

**Rating:** 4
**Confidence:** 3

**Review:**

The paper presents an algorithm and interface to retrieve parts of executions of an AI agent that comply with some specifications on the initial and final states. The paper describes a Linear Temporal Logic on Finite Traces (LTLf) that is used to formalize the queries. The query answering algorithm then builds NFAs on the input query represented as an LTLf formula and tests the input traces on it to find subsets of it accepted by the automaton. The authors evaluate the interface with a user study.

I found the paper to be interesting, and the running example was very useful to better understand the LTLf parts. However, I believe the contributions of the paper must be made clearer from the scope of the submitting venue. I would suggest making clearer what is the intent of the interface, and how are authors expected to obtain explanations from it. As this reviewer understood it, the users may request to see sub traces where some conditions happened, rather than asking questions on the actual behavior. The authors should make more clear how this links with actual explanations to the user, or what is the actual intent of the defined queries. They acknowledge that in the results, where they do not evaluate the explanations but the interface to query execution traces. I still find it interesting and a nice piece of work, and the proposed formalization looks sound, but the proposed tool seems more to provide a "recall" or "summarization" of past executions rather than an explanation.

Some questions and comments:
* Is there any reason for using an NFA vs DFA? That's not explained, and Table1 mentions converting to DFA, so it is a bit confusing. I know NFAs can be expressed as a DFA, but this should be clear.
* In figure 3's caption, what is the "Bot"?
* From the algorithm, it is unclear if it works on a single trace or if the algorithm is iterated through all the available traces, or if all traces are concatenated on a single trace.
* Is the proposed LTLf a contribution to the work, or what's described is a description of an existing LTL extension? I
* How is the NFA generation performed? Is that using a method from Giacomo and Vardi 2013, or something else?
* Was there significance in the results obtained? If so it should say.
* In Abstracting the MDP, P is not defined.
* in Linear Temporal Logic on Finite traces $\phi \wedge \phi$ maybe should be $\phi_1 \wedge \phi_2$? (as $\phi \wedge \phi$ results in $\phi$). Similarly in the end of the section, in the second bullet point, it says $\phi_1 \wedge \phi_1$.
* In Stays constant, should be between $\phi_s$ and $\phi_e$. In the next formula, e is undefined and unclear what it means (I guess it's the end of the trace?).
* The text in changes is exactly the same as in stays constant.
* Typo: in results "int the task" -> "in the task"
* In participants, it mentions 20 were female, it is unclear if all the rest were self-identified as male.
* Third paragraph of the second column of page 1, Jacobs et al. should be parenthesised.

---

### Meta-Review · Program_Chairs · 2022-04-30

**Recommendation:** Accept
**Confidence:** 4

**Metareview:**

The problem looks at the problem of developing an interface and rrelated algorithm to retrieve relvant agent behavior. The queries here are mapped to LTL_f formula which is then used to drive the retrieval process. Both reviewers found the paper interesting and well written. Reviewer xqkp brings up a good point that the method described here may be better termed as a summarization or recall process than an actual explanation. However I won't take this as a huge downside as I still believe these problems still falls within the interest of this workshop. Reviewer nZ9J also provides a lot of useful suggestions for improving the user study, which I would definitely recommend considering. All in all, I am happy to suggest the acceptance of this paper.

---

### Decision · Program_Chairs · 2022-04-30

Accept